# The Role of Lumen Apposing Metal Stents in the Palliation of Distal Malignant Biliary Distal Obstruction

**DOI:** 10.3390/cancers15102730

**Published:** 2023-05-12

**Authors:** Theodor Alexandru Voiosu, Mihai Rimbaș, Alberto Larghi

**Affiliations:** 1Carol Davila Faculty of Medicine, 021155 Bucharest, Romania; theodor.voiosu@gmail.com (T.A.V.); mrimbas@gmail.com (M.R.); 2Gastroenterology Department, Colentina Clinical Hospital, 020125 Bucharest, Romania; 3Digestive Endoscopy Unit, Scientific Institute for Research, Hospitalization, and Health Care, Fondazione Policlinico Universitario A. Gemelli, Catholic University, 00168 Rome, Italy

**Keywords:** lumen apposing metal stents (LAMS), biliary obstruction, endoscopic ultrasonography

## Abstract

**Simple Summary:**

Obstruction of the distal bile duct caused by various tumors arising in the region of the bile tract and pancreas has been traditionally treated by surgical or endoscopic therapy. However, in several cases, either surgical or endoscopic therapy via the conventional approach is not technically feasible. Recent advances in endoscopy techniques allow safe and effective relief of obstruction using a novel endoscopic device–the lumen apposing metal stent–which can be delivered using an endoscopic ultrasonography-guided approach. Our review analyzes the data on efficacy and safety of this new technique, compared to the already available therapeutic methods.

**Abstract:**

Malignant biliary obstruction (DMBO) has been traditionally managed by endoscopic retrograde cholangiopancreatography (ERCP). In the case of ERC failure, percutaneous transhepatic biliary drainage (PT-BD) has been widely utilized as a salvage procedure. However, over the last decade, endoscopic ultrasound-guided biliary drainage (EUS-BD) has gained increasing popularity, especially after the advent of electrocautery-enhanced lumen apposing metal stent devices (EC-LAMSs) which enable a one-step procedure, granting prevention of biliary leakage and minimizing occurrence of adverse events (AEs). In parallel, increasing evidence suggests a possible role of EUS-BD in the management of DMBO as a primary palliative drainage modality. In the current paper, we aim to review all the available evidence on the role of EUS-BD performed with EC-LAMSs and discuss salient technical aspects of this type of procedure.

## 1. Introduction

Distal malignant biliary obstruction (DMBO) is mainly determined by either pancreatic ductal adenocarcinoma, or, less frequently, by distal cholangiocarcinoma or ampulloma. All these entities carry a dismal prognosis, with an overall five-year survival of around 11% [1]. In fact, in most cases, even histopathologically, it is very difficult or even impossible to differentiate a pancreatic from a distal biliary duct cancer. This differentiation, however, is of little relevance in thecase of patients fit for surgery, which in the case of pancreatic cancer represents less than 20% of cases [2]. In up to 75% of these patients, jaundice is among the presenting signs, and is related to the obstruction of the distal bile duct [3]. Thus, biliary drainage is essential for patients who are not candidates for surgical resection in order to make possible administration of palliative chemotherapy. Preoperative drainage is also indicated when there is an indication for neo-adjuvant chemotherapy, or when the curative resection is delayed for various reasons [4].

Currently, endoscopic retrograde cholangiopancreatography (ERCP) is the standard of care in achieving biliary drainage and is performed in this setting as a primary intent. Percutaneous transhepatic biliary drainage (PTBD) is regarded in most centers as a second-line drainage option in cases where ERCP fails or is not technically feasible. Owing to significant technological advancements, the endoscopic ultrasound (EUS)-guided approach became an alternative to PTBD in the last two decades [5], and in experienced centers, European Society of Gastrointestinal Endoscopy (ESGE) has recently stated that it should be utilized instead of the percutaneous one to achieve biliary drainage after the failure of ERCP [6]. EUS-guided choledochoduodenostomy (EUS-CDS) and EUS-guided hepaticogastrostomy (EUS-HGS) are the two main techniques for biliary drainage in this setting. The advent of cautery-enhanced devices carrying small caliber lumen-apposing metal stents for EUS-CDS (LAMS) and made it possible to access the common bile duct at a site above from the tumor, either from the distal stomach or duodenal bulb in a one-step procedure, without the need for device exchange, with excellent technical and clinical success rates of the procedure.

The present paper will review the published data related to the use of EUS-guided LAMS for achieving bile duct drainage in patients with malignant distal bile duct obstruction.

## 2. Technical Aspects

The technique of EUS-guided biliary drainage (EUS-BD) has evolved significantly over the past decade, particularly since the first clinical case of LAMS was described in 2014 [7]. In short, after the window of bile duct approach from the duodenal bulb or, less frequently, the distal stomach is identified under EUS guidance, and the cautery-enhanced device is slowly advanced into the bile duct above the level of the tumoral obstruction using pure-cut current. After penetration and proper advancement into the bile duct, the distal flange is deployed under endosonographic real-time guidance, followed by device retraction for achieving apposition of the bile duct and gastrointestinal tract, when the proximal flange of the LAMS can be safely deployed (Figure 1, Figure 2 and Figure 3). This revolutionary type of device has brought a paradigm shift in the field of interventional endoscopy, but many debates about the technical aspects of this procedure are still unresolved. We will strive to provide a pragmatic overview of the main points of contention among endoscopists performing this technique.

### 2.1. Why Prefer EUS-CDS over EUS-HGS?

When comparing EUS-CDS with EUS-HGS, most data come from retrospective observational studies, reporting comparable technical and clinical success rates between these two techniques. Data from studies directly comparing these two techniques (in 329 and 303 patients, respectively) found similar technical and clinical success rates (94.7% and 88.4%% for EUS-CDS vs. 96% and 87.5% for EUS-HGS) [8,9], and no statistically significant differences in AE rates related to the procedures [10]. In a meta-analysis of 12 studies with 693 patients included, both these techniques had similar technical and clinical success rates. Of note, shorter procedural time and fewer early AEs were observed for EUS-CDS [11]. In a recent multicenter study, where 12 centers were involved, including 182 patients, and reporting 14 years of followup [12], a higher need for repeat stenting 30 days post-procedure was observed in patients who underwent EUS-HGS vs. those with EUS-CDS.

Concordant with the data presented above, the ESGE guideline recommended performing EUS-CDS over EUS-HGS in the setting of MBDO due to the lower rates of AEs with the former [6]. The recommendation was, however, rated as weak and of low quality. Thus, the choice of the EUS-BD approach in these patients usually relies on the preference and expertise of the endoscopist performing the procedure. Generally, we consider EUS-CDS to be easier to accomplish, since it usually does not require guidewire manipulation or device exchange over the wire. However, in patients with altered anatomic or concomitant gastric outlet obstruction, a EUS-HGS approach may be favored. Conversely, the presence of massive left liver lobe metastases usually represents a straightforward indication for EUS-CDS.

### 2.2. Hot or Cold?

Although, in principle, the dog-bone design of both the Axios and Spaxus types of LAMS allows for insertion without the use of electrocautery (EC), resulting in a “cold” rather than a “hot” device, the advantages of using a ”hot” approach have not really been contested among endoscopists, with studies showing a significant reduction in procedural times in the ”hot” approach [13]. Due to the fact that the ”hot” approach usually means a one-step procedure, without the need for the use of different accessories for bile duct puncture and dilation, including critical over-the-guidewire steps of device exchange, it is of no wonder that most of the endoscopists prefer using the ”hot” device. Moreover, in experienced hands, this approach can be performed even without the need for X-ray guidance [14], and even at the bedside of severely ill patients admitted in intensive care units [15].

### 2.3. Which Stent Size?

In the case of LAMS, when discussing stent size, the endoscopist should consider that this concept involves the diameter of the stent, the length of the stent, and the diameter of the flanges, all of which impact on technical and clinical outcomes, with important differences between various types of LAMS. For example, the HotAxios system is currently available in six different sizes, ranging from a diameter of 6 mm to a maximum of 20 mm and lengths from 8 to 15 mm, with a flange size ranging from 14 to 29 mm [16] while the HotSpaxus system has a fixed length of 2 cm and a variable diameter of the stent (8/10/16 mm) and flanges (23/25/31 mm).

The main issue that dictates stent choice in EUS BD is the size of the CBD, which is usually dilated at around 12–15 mm, with most experts feeling confident in performing EUS-BD with LAMS at a CBD size of 12 mm and above, as demonstrated in a large, multicentric trial evaluating the potential for EUS-BD by expert and non-expert endoscopists [17]. Consequently, smaller stent sizes (i.e., 6 or 8 mm HotAxios models, or 8 mm Hot Spaxus) are usually preferred [18,19] for EUS-BD. Of note, a non-negligible percentage of patients with DMBO included in this study did not meet the 12 mm threshold usually recommended for LAMS placement, suggesting that EUS-BD with LAMS might not be feasible in all cases of DMBO.

### 2.4. Combined EUS and X-ray, or Ultrasound Guidance Only?

Conventional wisdom suggests that X-ray guidance should be, at least, available in the endoscopy room, even if deployment of LAMS can be conducted using ultrasound and endoscopic guidance only. In fact, studies focusing on PFC have demonstrated excellent success rates for LAMS placement without fluoroscopy [20]. However, in the setting of EUS-BD, the technical challenges posed by smaller target sites [21] and unstable position of the scope, coupled with the potential for severe AEs in case of maldeployment, make the availability of fluoroscopy mandatory in this setting either to perform EUS-BD, or to be present in case of intra-procedural AEs.

### 2.5. Wire-Guided or Freehand?

The use of a hydrophilic guidewire (0.035 or 0.025”) is closely related to the use of fluoroscopy guidance discussed above, as well as to the expertise and confidence of the endoscopist performing the procedure. While most experts can safely perform EUS-BD in DMBO using a free-hand technique, a preloaded guidewire mounted on the LAMS catheter should be available in some clinical settings, such as minimally dilated bile ducts (<15 mm), unstable scope position, and in the case of less experienced operators. The use of a guidewire as an additional precaution has been shown to allow successful management of LAMS maldeployment in this clinical scenario, allowing salvage SEMS deployment through the newly created fistula tract in those cases where a LAMS was incorrectly deployed [21].

### 2.6. Coaxial Plastic Stent or Just LAMS?

This debate stems from the initial LAMS clinical application, which was the drainage of peripancreatic fluid collections. In this setting, long-term indwelling LAMS were associated with an increased risk of late-onset adverse events [22], with delayed bleeding in particular potentially linked to the traumatic effect of the metal stent. As a result, some experts advocated the use of a coaxial DPT in this setting to prevent delayed AEs. Building on the experience gained with the management of peripancreatic collections, coaxial DPT plastic stent placement was proposed for EUS-BD using LAMS. However, a recent study comparing placement of LAMS alone to placement of LAMS plus a coaxial DPT plastic stent showed no difference in terms of success rate or AE rate, with a significant longer procedure time in the coaxial stent group [23]. A Spanish multicenter randomized controlled study is ongoing and should provide a more definitive answer to this important question [24].

## 3. Current Evidence for EUS-BD Using LAMS

### 3.1. EUS BD Drainage after Failed ERCP

A large retrospective analysis of 256 patients showed very high rates of technical and clinical success in patients treated with EUS-BD with LAMS after failed ERCP [25]. The transduodenal access from the bulbus was used in more than 90% of the cases and Hot Axios of 8 × 8 mm or 6 × 8 mm were the preferred stents, used in 51.6% and 33.6% of the cases, respectively. Most procedures were performed in the same session after a failed ERCP (68.8%) and an overwhelming majority were conducted without fluoroscopy assistance (89%). However, only three cases of NagiStent and no HotSpaxus were included in this analysis which limits the generalizability of these results to stents other than the Hot-Axios stent. With regard to the HotSpaxus device, a single-center retrospective study showed similar clinical (93.3%) and technical success (100%) rates when using the 8 × 20 mm stents in patients with previously failed ERCP attempts. A recent systematic review of EUS-BD using LAMS [26] in 284 patients showed a 95% clinical success rate, with a low rate of adverse events (5.6%) and a rate of recurrent jaundice of 11.3%. Of note, most cases included in this analysis were treated using EC-LAMS (HotAxios), with only a few cases of “cold” LAMS or the HotSpaxus model (26 patients, i.e., less than 10%). A paper from France analyzing risk factors for stent dysfunction over a long-term follow up in 123 patients found that after a mean follow up of 242 days, 20 patients (16.3%) demonstrated biliary obstruction with successful desobstruction possible in 16 of them (80%). At uni- and multivariate analysis the factors associated with stent dysfunction were the presence of a duodenal stent and a bile duct diameter less than 15 mm [27].

When comparing EUS-CD with LAMS versus SEMS, a recent retrospective study showed very similar and high technical and clinical success rates in both groups, with a non-negligible rate of procedure-related AE rates but no statistical difference between the two methods [28]. These findings are in line with a recent meta-analysis including 820 patients treated by EUS-BD with either LAMS or SEMS, which showed comparable efficacy and safety profiles between the two groups [29].

### 3.2. EUS BD Drainage vs. ERCP as Primary Drainage

In light of recent data showing excellent clinical success and good safety profiles for LAMS as a second-line therapy in DMBO, EUS-BD has been recently proposed as a first-line alternative to ERCP for the management of DMBO [30]. Although data for a first-line EUS approach remains limited, a recent meta-analysis has shown that primary EUS-BD, including also EUS-guided hepaticogastrostomy (EUS-HGS), resulting in comparable clinical and technical success rates with primary ERCP, with no increase in AE rates but with an advantage in terms of patients undergoing chemotherapy for the EUS group [31]. Furthermore, a recent study has shown that EUS-BD using LAMS can also be successfully used as a bridge to surgery in selected cases [32]. At this time, we feel that EUS-BD as a first-line treatment for MBO should be reserved for the research setting and in selected cases where primary ERCP cannot be offered (i.e., modified anatomy, or documented duodenal obstruction).

### 3.3. EUS-Guided Gallbladder Drainage

EUS-guided gallbladder drainage (GBD) is usually reserved for patients where both ERCP and EUS-BD are not feasible or both failed; EUS-BD is usually hampered because of the presence of duodenal stenosis, thickened bile duct walls, intervening vessels, and/or nondilation of the intrahepatic bile ducts, as reported in a recent study [33]. In the most recent paper on this topic by Issa D et al. [34], the author described 28 patients in 26 of whom LAMSs were utilized over a six-year period with technical and clinical success rates of 100% and 93%, respectively. Adverse events occurred in three patients with food obstruction, two who developed cholecystitis, and one bleeding.

A recent meta-analysis of five studies looking at EUS-GBD as a rescue procedure after failed ERCP and EUS-BD showed good results, with pooled rates (95% CI) of clinical successm AEs of 85% (76%, 91%) and 13% (7%, 21%), respectively, and a 9% reintervention rate for stent dysfunction. Notably, four out of these five studies used LAMS for EUS GBD, with only one study using SEMS [35].

This approach should at present remain utilized only in those cases where both ERCP and EUS-BD fail and EUS proves the cystic duct to be patent. Interestingly, a recent study suggested the potential for EUS-GBD performed with LAMS as a prophylactic measure in patients receiving a conventional SEMS via ERCP to prevent post-ERCP cholecystitis [36], but the study has significant limitations, and the results should be interpreted with caution [37].

### 3.4. EUS vs. PTBD

PTBD is a simple and effective way to palliate MBO, both as a rescue therapy after failed ERCP and with a primary intention, especially in frail patients with complex hilar strictures [38]. However, the role of PTBD as the primary alternative to ERCP has been challenged by the advent of EUS-BD. A meta-analysis including six studies comparing EUS-BD with PTBD showed that EUS-BD was associated with similar clinical success rates but significantly fewer AE rates and reintervention rates [39] and these results were also confirmed by a systematic review which compared EUS-BD with PTBD in patients who failed an initial attempt at ERCP [40]. In light of this data, we believe that, whenever feasible, EUS-BD should be preferred over PTBD in DMBO cases where conventional ERCP drainage fails because of high efficacy and better safety profiles.

### 3.5. Altered Anatomy

Post-surgical anatomy represents one of the main challenges for ERCP, resulting in low clinical and technical success rates and a significant rate of severe AEs, especially perforations caused by the challenges of navigating the altered anatomy with a side-viewing endoscope [41]. As such, EUS-BD is a very good alternative to the transpapillary approach to endotherapy in cases of altered anatomy, since the bile duct can be more easily approached with an ecoendoscope in most cases. A meta-analysis showed good technical and clinical success rates for EUS-BD in patients with altered anatomy, with an overall rate of AE of 12.8% [42].

### 3.6. Surgery and EUS-BD

Surgical management of MBO usually includes choledoco-duodenostomy or choledoco-jejunostomy procedures which are highly effective and safe procedures in most patients fit enough to undergo surgery. A network meta-analysis of biliary drainage after failed ERCP showed that surgery was not superior to either EUS-CDS or EUS-HGS in terms of clinical success or safety, but this meta-analysis did not include studies from the LAMS era [43].

The issue of performing resection surgery with curative intent after EUS-BD in patients with MBDO is also a matter of concern. In an international, multicenter, retrospective study comparing patients who underwent hepatobiliary surgery after biliary drainage achieved by means of ERCP or EUS-BD (87 vs. 58 patients), the outcomes of the surgical resection were significantly better for the post-EUS-BD patients vs. the post-ERCP group (technical success 97% vs. 83%, clinical success 97% vs. 75%, AE rates 17% vs. 26%, total length of hospital stay 10 vs. 19 days) [44]. Thus, EUS-BD seems to be safe when utilized in DMBO prior to resection surgery. We can also speculate that EUS-BD can become the first treatment option in these patients, given the shorter duration between the endoscopic drainage procedure and surgical intervention, higher technical and clinical rates of surgical success, and shorter hospital stay after surgery when compared to those who underwent ERCP, as reported in the above-mentioned study. However, these findings need to be replicated in adequately powered multicenter prospective studies before widespread adoption of this approach can be advocated.

## 4. Ongoing Studies

In the field of EUS-guided choledochoduodenostomy, a couple of ongoing studies might provide more answers in areas of interest. In the PUMa prospective European multicenter trial [45], the investigators aim to prospectively enroll a target number of 212 patients in order to help clarify which of the two alternative biliary drainage techniques (PTBD or EUS-BD using a metal stent for patients with DMBO) has superior technical success and clinical efficacy rates as well as a better safety profile in patients who failed ERCP.

In the ongoing BAMPI Spanish trial [24], the investigators designed a multicentric open-label randomized controlled trial, where two parallel groups are compared after successful EUS-CDS with LAMS, with or without insertion of an axis-orienting (coaxial) double-pigtail plastic stent (DPS) through the biliary LAMS, in order to clarify if this additional maneuver is superior to LAMS alone for preventing stent dysfunction.

Last but not least, in the ELEMENT trial [46], a randomized multicenter single-blinded investigation aiming to enroll 130 patients with unresectable, locally advanced, or borderline resectable DMBO in nine Canadian centers. The study will include two arms: first intent EUS-CDS with LAMS vs. ERCP with traditional biliary metal stent insertion. The primary outcome is the rate of stent dysfunction needing re-intervention, and as secondary outcomes, technical and clinical success, delay or interruptions in chemotherapy, AEs, rate of surgical resection, and time to stent dysfunction. The study has been designed to assess whether EUS-CDS using a dedicated small-size LAMS is superior to conventional ERCP as a first intent drainage approach in patients with DMBO.

All these trials are intended to offer important and timely answers in areas where available data come mostly from retrospective or small prospective studies, thus, more accurate data from a multicentric randomized adequately powered trial are warranted.

## 5. Future Directions

In our opinion, the direction of development for EUS-BD will be to challenge standard ERCP as a primary biliary drainage approach for DMBO, independent of the endoscopic accessibility to the papilla. This should be a paradigm shift driven by the technical advancements of dedicated accessories to EUS-guided drainage to achieve easier and straightforward drainage of the biliary system but will also depend on data from cost-analysis studies which are, currently, still lacking.

Another potential benefit for patients is the possibility of a ‘one stop-shop’ approach to tissue acquisition and biliary drainage, which, when needed, could be accomplished with the same echoendoscope in one single endoscopic session [47].

In regards to training, there is also an urgent need for standardization, i.e., to devise the number of EUS-BD procedures required in training, along with standard ERCP training requirements, in order to be able to perform unsupervised drainage of the biliary ducts as an independent therapeutic endosonographer. The issue is important, given EUS-BD technical complexity and its potential for AEs which can be severe.

## 6. Conclusions

EUS-CDS using dedicated LAMSs has evolved to the point of offering effective and safe biliary drainage in patients with DMBO, with possibly a prolonged patency compared to the standard ERCP approach due to stent placement above and not through the tumor. The available evidence reports similar efficacy when compared to the standard ERCP approach, and superior efficacy when compared to PTBD in cases of failed or impossible ERCP. The rates and severity of AEs seem more favorable with EUS-CDS using dedicated LAMSs than with ERCP or PTBD, but the possibility of severe AEs still exists. However, further development of this technique needs to be coupled with ongoing advances in the field of echoendoscope design and EUS-dedicated accessories, and an improved training program. If successful, this could result in EUS-BD challenging ERCP as the standard for patients with DMBO.

## Figures and Tables

**Figure 1 cancers-15-02730-f001:**
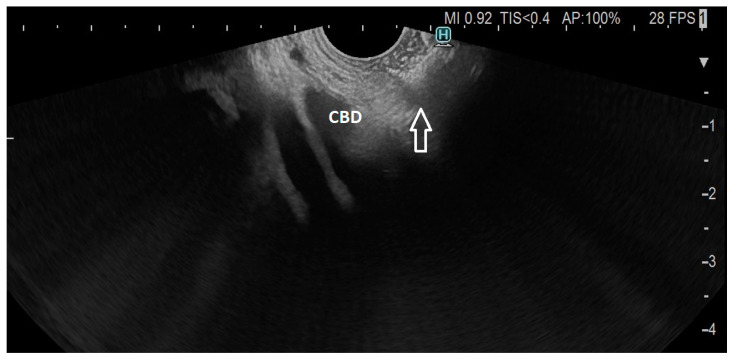
Dilated common bile duct (CBD) with EC-LAMS in the duodenum (arrow).

**Figure 2 cancers-15-02730-f002:**
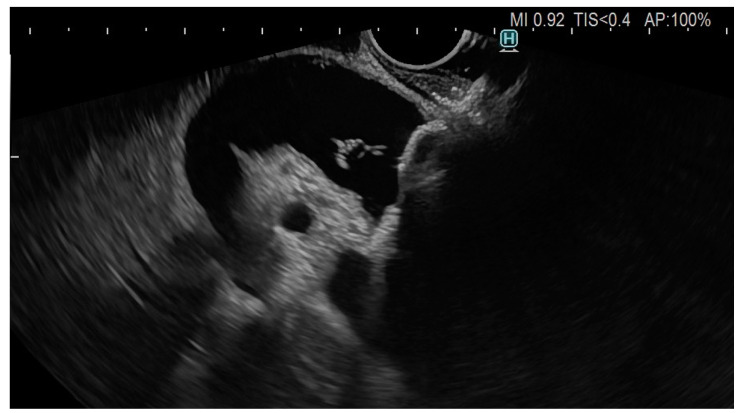
EC-LAMS inside the CBD, partial deployment of distal flange of the stent.

**Figure 3 cancers-15-02730-f003:**
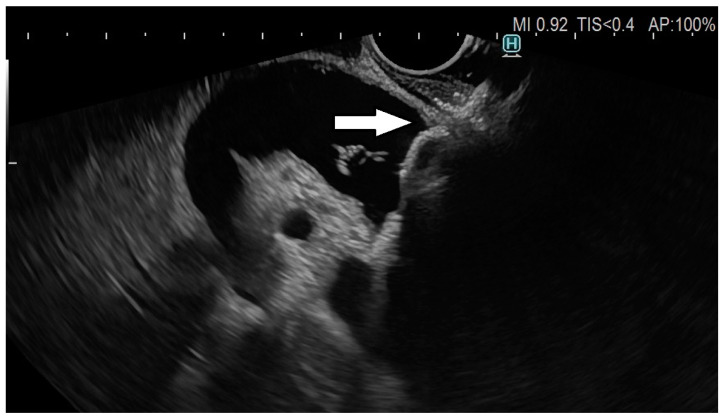
Fully deployed LAMS (arrow).

## Data Availability

The data can be shared up on request.

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
