# Peer review of "The Role of Lumen Apposing Metal Stents in the Palliation of Distal Malignant Biliary Distal Obstruction"

_cancers, 2023, doi:10.3390/cancers15102730_

Round 1

Reviewer 1 Report

well done and interesting paper to read and to know about

Author Response

thank you for the kind review

Reviewer 2 Report

Voiosu et al aimed to review all the available evidence on the role of EUS-BD performed with EC-LAMSs and discuss salient technical aspects of this type of procedure. The review is well structured and clear. No specific comment from my point of view.  

Author Response

thank you for the kind review

Reviewer 3 Report

EUS-CDS with LAMS for distal malignant bile duct obstruction has advantages indrainage, but whether it outperforms ERCP capable of diagnosis and treatment requires further discussion. On the other hand, EUS-HGS is one of the promising treatments for malignant bile duct obstruction with duodenal stenosis and postoperative intestinal reconstruction. Please add these points.

EUS-CDS with LAMS for distal malignant bile duct obstruction has advantages indrainage, but whether it outperforms ERCP capable of diagnosis and treatment requires further discussion. On the other hand, EUS-HGS is one of the promising treatments for malignant bile duct obstruction with duodenal stenosis and postoperative intestinal reconstruction. Please add these points.

Author Response

thank the reviewer for the kind suggestions, we included a segment on EUS HGS as an important alternative to EUSBD with LAMS in the section on altered anatomy. 

Reviewer 4 Report

I am also attaching the PDF text of the article with the corrections inserted in the notes to which I would rather add that the Bibliography needs to be reviewed as in some references there is the DOI  in others it is not!

Moderate editing of English language

Author Response

thank you for the suggestions, we have edited the references and the manuscript accordingly.